# Liquid Biopsy in Glioblastoma: Opportunities, Applications and Challenges

**DOI:** 10.3390/cancers11070950

**Published:** 2019-07-05

**Authors:** Ander Saenz-Antoñanzas, Jaione Auzmendi-Iriarte, Estefania Carrasco-Garcia, Leire Moreno-Cugnon, Irune Ruiz, Jorge Villanua, Larraitz Egaña, David Otaegui, Nicolás Samprón, Ander Matheu

**Affiliations:** 1Cellular Oncology group, Biodonostia Health Research Institute, E-20014 San Sebastian, Spain; 2CIBER de Fragilidad y Envejecimiento Saludable (CIBERfes), 28029 Madrid, Spain; 3Donostia Universitary Hospital, 20014 San Sebastian, Spain; 4Multiple Sclerosis group, Biodonostia Health Research Institute, 20014 San Sebastian, Spain; 5Spanish Network of Multiple Sclerosis, 08028 Barcelona, Spain; 6IKERBASQUE, Basque Foundation for Science, 48013 Bilbao, Spain

**Keywords:** liquid biopsy, glioblastoma, ctDNA

## Abstract

Liquid biopsy represents a minimally invasive procedure that can provide similar information from body fluids to what is usually obtained from a tissue biopsy sample. Its implementation in the clinical setting might significantly renew the field of medical oncology, facilitating the introduction of the concepts of precision medicine and patient-tailored therapies. These advances may be useful in the diagnosis of brain tumors that currently require surgery for tissue collection, or to perform genetic tumor profiling for disease classification and guidance of therapy. In this review, we will summarize the most recent advances and putative applications of liquid biopsy in glioblastoma, the most common and malignant adult brain tumor. Moreover, we will discuss the remaining challenges and hurdles in terms of technology and biology for its clinical application.

## 1. Glioblastoma

Glioblastoma is the most frequent, malignant and lethal primary brain tumor in adults. The World Health Organization (WHO) has classified it as a grade IV astrocytoma, the highest grade of malignancy, according to the histopathological, molecular and clinical features of these tumors [1]. Importantly, glioblastoma represents approximately 15% of all brain tumors and accounts for 50% of gliomas, with its incidence ranging from 1 to 5 cases per 100,000 individuals per year [2]. It presents an average survival of only 12–15 months from the time of diagnosis, standing as a highly aggressive tumor with an associated 5-year survival of less than 5%. The current therapeutic approach is maximal surgical resection, followed by concomitant radiotherapy and chemotherapy with temozolomide (TMZ) [3,4]. However, glioblastoma’s markedly proliferative, heterogeneous and chemoresistant nature highly compromises the available therapeutic options, leading to recurrence and death [1].

High-throughput large-scale sequencing analyses by The Cancer Genome Atlas (TCGA) have allowed a deep understanding of the molecular landscape of glioblastoma and have identified commonly deregulated genes and pathways such as growth factor signaling (RTK/Ras/PI3K), p53 and Rb signaling pathways [5,6]. These genetic and transcriptomic profiling analyses established a molecular classification comprising four subtypes: classical, mesenchymal, proneural and neural, although the latter was not clearly defined [7]. Thus, classical, mesenchymal and proneural subtypes are each mainly defined by aberrations in the gene expression of *EGFR, NF1* and *PDGFRA/IDH1*, respectively [7]. The distinct subclasses differ in pathological features as well as in clinical characteristics and are associated with different therapy response. Indeed, classical and mesenchymal glioblastomas present improved survival rates that correlate with their better response to conventional therapy, whereas this therapy does not improve the survival in patients with the proneural subtype, although this group presents the best outcome [7]. Such an outcome is explained by the global pattern of hypermethylation, referred to as the glioma-CpG island methylator phenotype or G-CIMP, which is closely related to the presence of *IDH1/2* mutations [8,9]. Nowadays, *IDH1/2* mutational status, rather than the subtypes described above, is the routine diagnostic molecular marker to differentiate among glioblastoma tumors, defining an *IDH* wild-type or mutant genotype [1]. In addition to the intertumoral heterogeneity, great heterogeneity also exists within each tumor due to the presence of different cellular populations or clones with distinct genetic or expression profiles. Among these, the glioma stem cell (GSC) subpopulation displays stem cell characteristics such as unlimited self-renewal capacity and multilineage differentiation potential, being responsible for tumor initiation and progression. Furthermore, since conventional therapies directly target proliferative cells, the undifferentiated and quiescent GSC population can lead to therapy resistance and frequently to tumor recurrence [10], all together forming a complex ecosystem that current models, aiming to study the biology of the tumor, are unable to recapitulate.

Overall, glioblastoma´s high heterogeneity at multiple levels (genomic, morphological, cellular, clinical and functional) hinders diagnosis and adequate therapeutic intervention, emphasizing the need to identify biomarkers for early diagnosis that allow a correct patient stratification and tailored therapy. Moreover, the dismal prognosis of glioblastoma highlights the need to unravel the critical molecular mechanisms underlying its progression to develop novel personalized therapeutic strategies. 

## 2. Use of Liquid Biopsy in Glioblastoma

At present, DNA and RNA are generally used to define a tumor’s molecular profile. However, the procedures for tumor tissue extraction are invasive, expensive and with risk to the patient. Moreover, tissue biopsies represent a small and localized region of the tumors and, therefore, might not fully capture the intratumoral heterogeneity. Furthermore, since the molecular landscape of the tumor evolves dynamically over time, biopsies might not be representative samples of the tumor, as patients cannot undergo such an invasive procedure repeatedly for the acquisition of samples. Recently, precision oncology and liquid biopsy have emerged as promising sources of biomarkers for diagnostic and prognostic purposes. In this regard, the analysis of biological body fluids has been shown to have the potential to determine the genomic profile of patients with cancer [11]. 

Glioblastomas are mainly diagnosed through neuroimaging techniques (e.g., magnetic resonance imaging (MRI) or computer tomography (CT) scans) and tissue samples, but they all have limitations [12]. For instance, MRI can only detect established tumors with sufficient mass. Moreover, the lack of accessibility to some brain tumors hampers obtaining tissue samples. The invasiveness of the procedure confers high risks to patients, such as brain swelling or hemorrhages that could result in deregulation of the brain function. Additionally, repeated surgery and sampling in order to define the real molecular profile of the tumor progression is not always possible. For these reasons, liquid biopsy appears as a promising approach to detect, molecularly characterize and monitor brain tumors and glioblastomas in particular [12]. A liquid biopsy might have some advantages versus tissue biopsy. It represents a simple and less invasive procedure that can provide similar information from certain body fluids (mainly blood) than what is usually obtained from a tissue biopsy sample [13]. It also has a potential clinical utility that could facilitate the early detection of cancer as well as improve patient follow up by managing tumor progression and monitoring therapy response [13,14]. In addition, this procedure could enable the identification of disrupted signaling pathways, the molecular subtype classification and also biomarker discovery. Importantly, liquid biopsies could be obtained regularly over time, reflecting the real composition, the tumoral heterogeneity and the evolution of the tumor throughout time.

Several biological analytes such as circulating tumor cells (CTCs), circulating cell-free DNA (cfDNA) that contains circulating tumor DNA (ctDNA) in cancer patients, circulating cell-free tumor RNA (ctDNA) containing mRNAs and mainly small RNAs, extracellular vesicles (EVs), proteins, metabolites and tumor-educated platelets (TEPs) are found in body fluids and could be sampled within a liquid biopsy (Figure 1). This circulating material can derive from tumor tissue and thus, can provide a real and representative sample of the glioblastoma. Indeed, the genomic profiles of ctDNA in liquid biopsies and the corresponding tumor have been found to closely match and correlate. Moreover, rapid turnover of glioblastoma cells could result in a constant release of tumor-derived nucleic acids and vesicles into circulation. Additionally, viable tumor cells could enter the bloodstream once separated from the neoplasm. In the following sections, we will describe the information obtained in liquid biopsies components derived from blood and cerebrospinal fluid (CSF) of glioblastoma patients. 

## 3. Liquid Biopsy Components from Blood

### 3.1. Cell-Free Nucleic Acids

Among the different types of materials that can be found in the blood of cancer patients, cell-free nucleic acids, ctDNA and ctRNA, are the most prevalent and well-studied components (Table 1). In particular, an increased amount of ctDNA has been found in blood samples from cancer patients, including glioblastoma, compared to control individuals or individuals with benign or low-grade tumors; indeed, the ctDNA level has been shown to positively correlate with tumor stage [12,15]. In glioblastoma, the presence of brain tumor-derived ctDNA in plasma is low compared to other cancers, mainly because of the presence of the blood–brain barrier (BBB), in contrast to other tumors that are able to transfer ctDNA fragments into blood [16,17]. For ctDNA analyses, plasma samples are preferable to serum ones because they represent a good source of ctDNA and lack background levels of cfDNA, which are higher in serum probably due to contamination with DNA released from immune cells lysed during the clotting process [18]. Nonetheless, plasma often contains low levels of ctDNA that vary between patients [16]. The exact mechanisms by which ctDNA is released into blood are still unclear; however, apoptosis and necroptosis are suggested as possible sources, since apoptosis is the principal source for both total cfDNA and ctDNA based on the size distribution analysis of cfDNA [19,20]. Additionally, macrophages seem to play a role in the liberation of tumor-derived DNA fragments into circulation via the phagocytosis of necrotic neoplastic cells. 

Since ctDNA quantification from plasma samples is challenging due to the low amount of biological material available [16], improvement in sequencing technologies is necessary. In this sense, the development of new technologies such as droplet-based digital PCR (ddPCR) [53] or the optimization of next generation sequencing (NGS) techniques [54] have improved the sensitivity and specificity for the detection of ctDNA mutations [55]. While NGS allows exploring a wide range of nucleic acids present in a sample with high sensitivity and reproducibility, with the use of ddPCR, the target sequences can be individually tested allowing rare event detection and quantification at the level of a single molecule. On the other hand, NGS can detect novel and unknown genetic modifications but longer time is needed to obtain, process or analyze the results; moreover, it requires bioinformatics expertise and it is more expensive [16]. In comparison with NGS, ddPCR experiments are easier to set up, faster, present higher sensitivity, lower costs and they do not need complex bioinformatic analysis. However, they only enable the study of known mutations [56]. In this sense, a technique comparison study comprising 62 glioblastoma samples to detect *IDH1* mutations by Sanger direct sequencing, ddPCR or quantitative real-time PCR (qRT-PCR) showed that ddPCR was more sensitive method to screen for *IDH1* mutations [57]. 

Few studies have analyzed and identified tumor-associated mutations in blood samples. An initial study detected *IDH1*, *EGFR*, *TP53* and *PTEN* mutations in a limited subset (10%) of patients with glioblastoma [16]. More recent studies performing comprehensive ctDNA analysis and a highly sensitive and specific NGS assay yielded approximately 50% of the ctDNA detection rate in patients with advanced glioblastoma and showed that the ctDNA detection rate in gliomas may vary by grade and histopathology [23,58]. A similar NGS analysis of plasma-derived ctDNA extracted from a total of 171 patients with different cancers, including 33 patients with glioblastoma, successfully determined the most frequent mutations in the *TP53, EGFR, MET, PIK3CA* and *NOTCH1* genes [59]. In other studies, the most frequent somatic mutations detected using NGS technology were observed in *TP53, NF1, EGFR1, MET, APC* and *PDGFRA* genes together with amplifications of *ERBB2*, *MET* and *EGFR,* among others [23]. These results should be confirmed by more sensitive techniques such as ddPCR or improved NGS; nevertheless, the spectrum of mutated genes is similar to that obtained by TCGA tissue analysis [5,6]. Thus, ctDNA might provide a comprehensive view of the glioblastoma genome, as it might reflect DNA derived from multiple tumor regions and could serve as a diagnostic biomarker. Since some of those mutations have been used to molecularly classify glioblastoma (*EGFR* amplification associated with classical, *NF1* and *TP53* mutations with mesenchymal or *IDH1* mutation with proneural subtypes), these results also suggest that plasma ctDNA analysis might be an option to obtain actionable somatic genomic information and reveal the molecular profile or subtypes of glioblastoma patients. This might also guide clinical therapeutic selection. Indeed, *BRAF/IDH1/IDH2* mutations, *PDGFRA* amplifications and mutations in DNA damage repair genes were proposed as potential candidates for molecular targeted therapy [23]. 

Moreover, epigenetic changes have also been detected in glioblastoma blood samples. Thus, methylation in promoter *MGMT* gene was observed in matched tissue and serum ctDNA from glioblastoma patients as well as correlated with clinical response to therapy [21]. Specifically, an improved response and time to progression after treatment with alkylating agents was reported in patients with increased serum levels of *MGMT* promoter methylation. Additionally, ctDNA from glioblastoma patients presented lower levels of Alu methylation than the controls [60], further supporting that ctDNA might provide a comprehensive view of the glioblastoma epigenome. A recent, sensitive immunoprecipitation-based protocol has been developed in order to study the methylome of small quantities of ctDNA to detect enrichment of tumor-specific patterns early in the progression of the disease [22]. It would be interesting to test this technique in future studies with glioblastoma samples.

Circulating cell-free tumor RNA (ctRNA) include mRNAs, long non-coding RNAs (lncRNAs) and mainly small ncRNAs. Small ncRNAs include in turn microRNAs (miRNAs), small interfering RNAs (siRNAs), circular RNAs (circRNAs), small nuclear RNAs (snRNAs) and small nucleolar RNAs (snoRNAs), which can be released into the extracellular medium to regulate several signaling pathways through repressing multiple genes at translational level. Among them, miRNAs have arisen as promising biomarkers for cancer diagnosis in the last decade [61]. These are small (18–23 nucleotides) endogenous ncRNAs involved in most physiological and pathological cellular events such as proliferation, differentiation, migration, invasion, senescence and survival by regulating post-transcriptional gene expression. miRNAs are the most abundant circulating free molecules in blood [62]. Moreover, detectable miRNA levels can be observed in additional cell-free body fluids as well as in tissues [63]. Thus, altered miRNA expression patterns in biological fluid samples correlate with tumor tissue samples and could improve early diagnosis, classification and prognosis prediction of different cancer types, including glioblastoma [63,64].

The expression of several miRNAs and lncRNAs has been reported as altered in glioblastoma patient samples [65,66]. In particular, a global serum miRNA signature in a large cohort of malignant glioma patients revealed that seven serum miRNA (miR-15b, miR-23a, miR-133a, miR-150, miR-197, miR-497 and miR-548b) levels were reduced in the serum of cancer patients compared to normal controls [24]. Significant upregulation of miR-21 and downregulation of miR-128 and miR-342 were also found in plasma and tissue samples from glioblastoma patients compared to healthy controls [24], postulating them as predictive biomarkers for diagnosis. Moreover, high-grade glioma patients presented lower miR-128 and miR-342 levels in plasma suggesting that these miRNAs positively correlate with histopathological grades of glioma [25]. Remarkably, their levels were restored after surgery and chemo-irradiation therapy, suggesting they could also be useful biomarkers for therapy response [25]. Interestingly, these miRNAs seem to be specific for gliomas, allowing the distinction between glioblastoma and other brain tumors, such as meningiomas or pituitary adenomas. Similarly, a restricted signature of serum miRNAs including miR-125b and miR-497 levels could distinguish high-grade and low-grade gliomas with a downregulation of these two miRNAs in glioblastoma patients compared to the lower grade group [26,27]. miR-205 represents another potential diagnostic biomarker, since it was found significantly decreased in serum from patients with glioblastoma, and its expression negatively correlated with the pathological grade of the tumor [28]. Of note, serum miR-205 levels increased after surgical resection of the tumor and dropped again upon recurrence, being a good candidate for tumor evolution monitoring. miRNAs can also be overexpressed in glioma samples. Specifically, miR-21expression was found highly expressed in glioblastoma patients and was described to inversely correlate with survival of those patients [25,29]. Furthermore, miR-21 could be a biomarker of therapeutic response since its levels diminished after chemo-radiation. Likewise, elevation of serum miR-221, miR-222, miR-210, miR-182 or miR-454 levels were observed in glioblastoma patients and also associated with tumor progression and low survival rates, which postulate them as good prognostic biomarkers [30,31,32,33]. These studies highlight the use of a panel comprising multiple miRNAs found in serum from glioblastoma patients as a potential strategy for diagnostic and prognostic purposes. However, the usefulness of miRNAs as biomarkers for disease can be controversial as different studies show distinct expression of the same miRNA in glioma cells, underscoring the careful selection that must be carried out in order to implement miRNAs as biomarkers for glioblastoma diagnosis in clinical practice [66]. Indeed, it has been proposed that miRNAs need to be validated in further research studies and clinical trials, as well as a standardization of sample processing, miRNA detection and statistical analysis methods should be established before implementing miRNAs as biomarkers in clinics [67].

LncRNAs are commonly involved in gene regulation, leading to several signaling pathway deregulation and ultimately promoting carcinogenesis [68]. Thus, they have emerged as potential diagnostic and prognostic biomarkers in glioblastoma [65]. Of note, lncRNA and miRNA functions are intimately correlated and indeed form a complex regulatory network that modulates each other’s expression. They are key molecules in mRNA stabilization and degradation, but they are also involved in the miRNA sponge mechanism influencing multiple biological processes (expression at a post-transcriptional level, epigenetic regulation, cell growth and death) [65]. Several lncRNAs have been detected in blood samples from glioblastoma patients with differentially expressed levels compared to healthy individuals. Among them, the epigenetic regulators HOTAIR, GAS5, H19 and MALAT1 were found in serum and tumor samples from glioblastoma patients [34,35,36]. Interestingly, GAS5 has been related to the responsiveness of glioblastoma patients to TMZ, having potential therapy response prediction value. Additional ctRNAs, including siRNAs, circRNAs, snRNAs and snoRNAs have been detected as differentially expressed in glioblastoma blood and tumor samples. This reveals their potential as biomarkers for diagnosis and prognosis as well as suggests a role in the pathogenesis of the disease [69,70]; however, additional experiments and validations are required to support their roles.

### 3.2. Extracellular Vesicles 

In recent years, EVs have boosted new opportunities for non-invasive diagnosis and monitoring of human cancer. EVs are small membrane-enclosed spheres (40 to >1000 nm in diameter) produced and secreted by many different cell types through complex and tightly regulated molecular mechanisms [71,72]. The different types of EVs are classified depending on their biogenesis into exosomes (40–200 nm), membrane-derived vesicles (40 to >1000 nm) and apoptotic bodies (100–5000 nm) [73]. EVs are present in several body fluids such as blood, CSF or urine, and they have a diverse molecular composition including nucleic acids, proteins, lipids and metabolites that can be transferred to nearby or distant cells by direct EV–cell membrane contact, fusion or internalization [72,73]. Unlike ctDNA, EVs arise from viable cancer cells; consequently, these two analytes might reflect different aspects of tumor biology. The relevance of EVs is highlighted by the fact that their transcriptomic and proteomic profile is specific depending on the cell of origin and can vary in response to diverse stimuli. Indeed, EVs can carry tumor-specific material that might serve as a diagnostic and prognostic tool [74]. In addition, they participate in the modulation of cancer hallmarks by transmitting oncogenic signals to recipient cells in order to promote tumorigenic activities such as proliferation, survival and differentiation [41,75,76]. In addition to their stability, EVs hold the capacity to protect and maintain the integrity of their content, preventing degradation and enabling its further study. For instance, EV-derived DNA may be representative of the mutational status of parental tumor cells and could, therefore, be of relevance for genomic analysis [77].

The analysis of EVs detected in blood, CSF or any other biological fluid from glioblastoma patients is gaining interest and they are emerging as candidates to be useful diagnostic and prognostic biomarkers [75]. Thus, it has been described that plasma EV concentration is higher in glioblastoma patients compared to healthy controls and that the proteomic profiling of these EVs revealed a glioblastoma-distinctive signature with *EGFR* amplifications, *PTEN* deletions, and *IDH1/2* and *TP53* mutations [41]. Noteworthy, EVs can transgress anatomical compartments such as the BBB and that raise them as a potential source of biomarkers for glioblastoma to detect evolving changes relative to tumor progression [42,43,78,79]. Indeed, molecular alterations in various genes such as *EGFRvIII*, *IDH1* and *PTEN* have been found in glioblastoma-derived EVs [43,78]. Moreover, EVs can mediate the crosstalk between glioblastoma and its microenvironment, exchanging signals between the brain cells and the surrounding stroma and altering the tumor milieu to provide a suitable environment for tumor growth [80]. In this sense, the interplay between EVs and anticancer immunity represents a novel area of research and potential biomarkers are starting to be identified [81]. PD-L1 expression has been found on the surface of glioblastoma-derived EVs [45]. These PD-L1 expressing EVs can prevent T cell activation and proliferation upon direct binding with PD-1, indicating that PD-L1 expression on EVs is an immune-escape mechanism for glioblastoma [45]. Interestingly, several studies have reported that TMZ treatment can affect EV release and confer drug resistance to recipient cells by horizontal transference of molecular cargos through EVs. For instance, EVs collected from TMZ-resistant patients showed increase expression levels of *MGMT* [44,46]. Moreover, TMZ-resistant cell-derived EVs present deregulated levels of cell adhesion-related proteins like transglutaminase 2 (TGM2), stemness markers such as NESTIN and glycoproteins such as CD44 and CD133 that are expressed in the surface of EVs [46,82]. This suggests that they could be exploited as cancer biomarkers to monitor TMZ failure through the analysis of their molecular components.

### 3.3. Circulating Tumor Cells 

CTCs are tumor-derived cells that have entered the bloodstream or have been passively released from the primary tumor into circulation [83]. Their abundance in blood is very limited with only 1 CTC per 10^9^ blood cells, and their isolation is difficult due to the complexity of the required techniques. Negative-enrichment approaches are based on CTC size (CTCs are larger than normal blood cells) or other biophysical properties. Positive selection of CTCs can be achieved by the detection of specific tumor markers that are commonly expressed on the surface of these cells [84]. Nevertheless, these surface markers are not specific, hampering the distinction of CTCs from non-malignant cells. So, DNA, RNA or protein analyses have to be performed to fully analyze the content and identify CTCs. There are still few studies that relate CTCs to clinical applications, but the improvement of new techniques, such as single-cell analysis, have allowed the shift from the enumeration of CTCs to a more detailed genomic, transcriptomic, proteomic and epigenomic analyses of their content. This is providing us with valuable information that promotes the use of CTCs for diagnostic, prognostic or therapeutic purposes [85,86]. Even though it is difficult to detect them, the abundance of CTCs in the blood of patients with cancer has been proposed to have clinical utility as a biomarker for prognosis and might also be useful for therapeutic assessment [84,87]. CTCs have been detected in the blood of patients with different grades of glioma including glioblastoma, and they have been shown to be representative of the tumor [48,49]. In line with this, glioblastoma-derived CTCs presented *EGFR* amplification that was linked to aggressiveness and was associated with the presence of *EGFRvIII* [49]. Moreover, the single-cell expression analysis of 15 CTCs from seven independent glioblastoma patients revealed an elevated expression of *SERPINE1, TGFB1, TGFBR2* and *VIM* genes associated with the mesenchymal subtype [47]. Additional studies have revealed that glioblastoma-derived CTCs possess stem cell properties contributing to local tumorigenesis and recurrence [50,51]. Indeed, cultured CTCs were described to express GSC markers including *SOX2*, *OCT4*, and *NANOG* and also present resistance to the treatment with radiotherapy or TMZ in comparison with cultured tissue-derived glioblastoma cells [50]. Despite being possible candidates for clinical applications in glioblastoma, the challenging isolation and the low number of CTCs detected restricts the number of available studies.

### 3.4. Proteins and Metabolites

Measurements of serum circulating proteins are currently being used in clinical practice to identify potential reliable biomarkers for cancer detection. Several studies have investigated specific serum-derived proteins for glioblastoma characterization. Thus, it has been reported that serum from patients with glioblastoma exhibited 27 differentially expressed proteins, of which five had been previously associated with the progression of the tumor [88]. Moreover, the expression of haptoglobin α2 was significantly elevated in glioma patients and positively correlated with tumor grade, being the highest levels detected in patients with glioblastoma [37]; this postulates haptoglobin as a candidate biomarker for glioblastoma diagnosis and prognosis. In contrast, chemo-irradiation did not alter the serum levels of haptoglobin α2 as they were only reduced after adjuvant therapy [89]. Glioblastoma biopsy samples and serum levels also displayed an elevated YKL-40 expression that was associated with worse prognosis and overall survival of patients [38,39]. Likewise, the serum AHSG concentration was found to be increased in glioblastoma patients and was associated with higher tumor grade and lower overall survival [40]. However, other studies have demonstrated no association between the serum levels of haptoglobin α2, YKL-40 or AHSG and progression-free survival [89]. 

Metabolism is critical in order to sustain cancer cell proliferation and adaptation to tough microenvironments. Thus, elucidating and understanding metabolic reprogramming is a key for biomarker discovery and development of therapeutic strategies. In this regard, circulating metabolite levels are interesting biomarker candidates for cancer detection and progression. So far, there are not many studies in glioblastoma but some of them support the use of some metabolites as biomarkers in liquid biopsies. Thus, metabolomic profiling identified arginine, methionine and kynurenate levels in plasma significantly associated with 2-year overall survival [90], revealing a metabolite profile predictive of glioblastoma prognosis. Moreover, an additional study revealed higher serum concentrations of α-tocopherol and γ-tocopherol in serum samples from glioblastoma, postulating them as latent biomarkers for glioblastoma progression [91].

### 3.5. Tumor-Educated Platelets 

TEPs are referred to as the platelets that have received tumor-associated molecules from neoplastic cells [92]. This biomolecule transfer process is known as “education”. TEPs have been shown to actively contribute to cancer growth and metastasis, providing angiogenic factors to support vascularization. Indeed, angiogenesis-related proteins can be detected in platelets before the tumors have reached detectable sizes, suggesting a potential avenue for the early and minimally invasive detection of cancer [93,94]. Thus, it has been proposed that RNA from TEPs may complement the currently used biosources and biomolecules employed for liquid biopsy diagnosis, which would potentially enhance the detection of cancer in an early stage and facilitate noninvasive disease monitoring [94]. TEP RNAseq analyses of 228 localized and advanced metastatic cancer patients, comprising various tumor types including glioblastoma, evidenced different RNA profiles between cancer patients and healthy individuals. Indeed, discrimination between cancer patients from healthy individuals could be achieved with 84–96% accuracy by analyzing TEPs, being glioblastoma cases within the most accurate ones. Moreover, TEP profiling might serve to establish the organ of origin of the primary tumor with 71% accuracy and indeed TEP profiles were shown to be distinct between different molecular tumor subtypes based on *EGFR* and *K-RAS* [95]. Furthermore, it has been described that TEPs can take up EV-derived RNA. In this regard, TEPs from glioblastoma patients were shown to capture tumor-derived EVs containing mutant *EGFRvIII*. Remarkably, *EGFRvIII* mutation was detected in 80% of glioblastomas but not in healthy controls [52]. This finding probably extends to other tumor-related mutations, as RNA profiling from glioblastoma and healthy patients led to the identification of a glioblastoma-associated signature. Nonetheless, the utility of TEPs as non-invasive biomarkers for cancer diagnosis and classification remains to be fully demonstrated, since the “education” process is still poorly understood and the platelet subpopulations could change during the clinical course of glioblastoma patients [95].

## 4. Liquid Biopsy in Cerebrospinal Fluid 

CSF is a colorless biological fluid produced by the specialized ependymal cells within the choroid plexuses of the brain ventricles. CSF is obtained through lumbar puncture, which is an invasive procedure that might be performed in patients harboring brain tumors, including glioblastoma. Since CSF is in direct contact with the central nervous system (CNS), liquid biopsy of CSF can be used to profile circulating biomarkers like cell-free nucleic acids (ctDNA and miRNAs), proteins, EVs or CTCs of glioblastoma patients (Table 2). Moreover, it could also be useful to detect circulating material from tumors that have metastasized to the brain. 

The identification of glioblastoma biomarkers in CSF was first achieved by the detection of high levels of tenascin, an extracellular matrix glycoprotein, in brain tumor patients compared to other non-CNS primary tumors and control patients [104]. Similarly, elevated levels of osteopontin (OPN) were noted in CSF samples from glioblastoma patients compared to healthy control CSF [105]. An additional study found that cleaved OPN fragments in CSF were more abundant in glioblastoma patients and, after the analysis of consecutive CSF samples, they identified that these fragment levels were positively correlated with CSF levels of vascular endothelial growth factor (VEGF) (marker of angiogenesis) and TNFα, CCL3 and CCL4 (indicators of inflammation) [106]. In this sense, the levels of VEGF and fibroblast growth factor 2 (FGF2) were shown to be significantly higher in patients with glioblastoma and associated with shorter overall survival rate, demonstrating the usefulness of these proteins as potential diagnostic and prognostic biomarkers [107]. Furthermore, the expression of the nerve growth factor (NGF) in CSF was found to be more elevated as the malignancy of the glioma increased [108]. More recently, a robust association between tumor type and tumor location with the presence of ctDNA in CSF has been described and correlated with disease burden, adverse outcome and shorter overall survival [17]. In this study, matched samples of CSF, plasma and tumor-derived tissue DNA from glioblastoma, medulloblastoma or brain metastasis were studied and highlighted the potential of CSF-derived ctDNA to characterize and monitor brain tumors in comparison to blood samples. Noteworthy, CNS malignancy-derived ctDNAs were more abundant in CSF than in plasma samples [17]. Similarly, a genome-wide sequencing study involving 35 patients with primary CNS tumors, could detect CSF ctDNA in 74% of the cases and in 100% of high-grade gliomas localized neighboring a CSF reservoir, whereas no ctDNA was noticed in patients with CNS tumors physically distant from this reservoir [110]. Thus, circulating components in CSF samples stand as an alternative source to blood to identify diagnostic and prognostic biomarkers in malignancies affecting the CNS. 

The study of De Mattos-Arruda and collaborators also found mutations in *IDH1/2*, *EGFR*, *PTEN*, *FGFR2* and *ERBB2* genes in CSF samples, and showed that CSF ctDNA is longitudinally modulated throughout treatments after the observation of dynamic changes in CSF ctDNA, which recapitulates the treatment courses of patients with glioblastoma [17]. Thus, ctDNA from CSF could be used to study somatic mutations and monitor tumor burden. In this regard, a recent study in CSF from 85 glioblastoma patients revealed that ctDNA was detected in 49.4% of patients and associated with disease burden and adverse outcome [96]. Moreover, the genomic landscape included a broad spectrum of genetic alterations that closely resembled the genomes of tissue biopsies including mutations in *TP53* and *IDH1/2*, deletions of *CDKN2A* and *CDKN2B*, and amplification of *EGFR* [96]. ctDNA mutations in *IDH1/2* were shared in all matched ctDNA-positive CSF-tumor pairs and further analysis of CSF ctDNA revealed a broad spectrum of promoter mutations, copy number alterations and structural rearrangements [96]. Sequencing analysis also detected somatic alterations in a panel of 341 cancer-associated genes in the CSF ctDNA from patients with primary brain tumor or metastases from solid tumors, but not from patients without CNS affection by cancer [111]. These results suggest that CSF better captures genomic alterations and that is more representative of glioblastoma genetic alterations than plasma ctDNA, recapitulating the mutations from tissue samples. Noteworthy, promoter hypermethylation in *MGMT*, *p16INK4a*, *TIMP-3* and *THBS1* was detected at high frequencies in CSF, serum and tumor tissue, in all glioblastoma patients but not in any of the healthy individuals [97]. Hypermethylation in CSF and serum DNA was accompanied with methylation in the corresponding tumor tissues with 100% specificity [97]. Moreover, hypermethylation of *MGMT* and *THBS1* in CSF were the only independent prognostic factors, indicating that CSF also captures epigenomic alterations. 

Additional components of liquid biopsy have been detected in CSF from glioblastoma patients. In this sense, EVs derived from CSF allowed the detection of *IDH1* mutations in five out of eight patients using BEAMing and ddPCR techniques with matched corresponding *IDH1* mutations in tumor tissue [109]. In addition, circulating miRNAs and EV-derived miRNAs were detected in CSF samples [98,99], and these results allowed to propose that they might serve as CSF biomarkers to diagnose and monitor therapy response in glioblastoma patients [99]. In support of this, the levels of a panel of nine miRNAs, including miR-21, miR-218, miR-193b, miR-331, miR374a, miR548c, miR520f, miR27b and miR-30b, in CSF from glioblastoma patients were found to correlate with those detected from tissue biopsies samples [100]. Such signature was associated with tumor volume and presented 67% sensitivity and 80% specificity [100]. Likewise, upregulated levels of miR-21, miR-10b and miR-15b in CSF have been associated with tumor stage, prognosis and response to therapy in glioblastoma patients [101,102,103]. Interestingly, disease progression and therapeutic response was reflected by the CSF levels of miR-21 in longitudinal samples of glioblastoma patients [102]. The presence of miR-21 together with mir-15b allowed differentiating glioblastoma from other intracranial pathologies with 90% sensitivity and 100% specificity [101]. Likewise, these CSF miRNAs have shown to have better diagnostic value, with high sensitivity (84%) and specificity (92%) than serum levels to identify patients with glioblastoma. Moreover, miR-21 was highly overexpressed in glioblastoma-derived EVs (tenfold higher levels) compared to EVs isolated from the CSF of healthy patients [82]. Interestingly, analyses of longitudinal samples reported a correlation between tumor burden and EV miR-21 levels in CSF, as miR-21 content decreased after surgery [82]. Together, these results imply the relevance of miRNA profiling in CSF for glioblastoma diagnosis. Nevertheless, as stated before there is still need for further standardization of the miRNA analysis, isolation and quantification methods. Importantly, these results highlight that CSF samples stand as a source to identify diagnostic and prognostic biomarkers. Moreover, ctDNA sequencing from CSF samples could facilitate glioblastoma genotyping and characterization and might complement the information provided by other non-specific biomarkers and the imaging techniques that are currently implemented in clinical practice.

## 5. Other Sources of Liquid Biopsy: Urine

Urine represents a useful source of biomarkers as its collection is non-invasive, simple and could be repeatedly done throughout the course of the disease. In fact, ctDNA can be detected in urine after renal filtration and is referred to as transrenal DNA (trDNA). This analyte appears in urine as a result of the renal clearance of the ctDNA present in the bloodstream [112]. A panel of urinary biomarkers, comprising matrix metalloproteinases (MMPs) and VEGF, was described to possess predictive ability to detect primary brain tumors including glioblastoma with high sensitivity (95.2%), specificity (95.7%) and overall accuracy (92.5%) [113]. Thus, the *MMP-2*, *MMP-9*, *NGAL* and *VEGF* markers were elevated in samples from glioblastoma patients compared to control individuals. The elevated expression in urine correlated with the increased expression of such biomarkers in the tumor tissue, as well as in the CSF in the case of *MMP-9* expression. However, the diagnostic ability of the biomarker panel in glioblastoma was reduced, since it was developed for a wide variety of brain tumors and not specifically for this tumor. Additional studies and clinical trials have used urine to measure the expression of biomarkers but without robust results [114,115,116,117].

## 6. Conclusions and Future Perspectives

Glioblastoma remains one of the most aggressive human cancers with current diagnosis and prognosis based on imaging and tissue sample information. There is therefore an urgent clinical need for the identification of non-invasive methods for diagnosis, monitoring the evolution and use of genotype-directed therapies. Liquid biopsy could offer a minimally invasive possibility to unveil the molecular landscape of solid tumor samples through the analysis of biological fluids from patients. 

Research in liquid biopsy is gaining remarkable interest in glioblastoma. Indeed, sampling body fluids such as blood or CSF derived from glioblastoma patients revealed that they present several circulating components that derive directly from the tumor. Thus, it might be an alternative source to tissue sample. These components include relatively well-established sources such as ctDNA, ctRNA, EVs and CTCs, or promising ones such as different metabolites and TEPs. They can have great potential to identify biomarkers for diagnosis, prognosis, and therapeutic response monitoring purposes, and could also allow genotype-directed therapies and the personalized clinical management of gliobastoma patients (Figure 2). 

Since glioblastomas arise in the CNS, accessibility is a problem and the BBB hampers the release or transfer of tumor-derived material into blood, which has been the most studied biological fluid until recently. Thus, serum or plasma components, in particular tumor-derived nucleic acids, have been detected generally at low levels and this might make its implementation difficult in routine clinical practice. This might happen because the levels are insufficient to perform genotyping and detect the somatic mutations of the tumor. In order to overcome these limitations, CSF is emerging as a promising sample and, together with improved sequencing approaches with increased analytical sensitivity and specificity, it has recently shown promising results towards ctDNA analysis [96,118]. New technologic advancements could allow for continued refinements in standardization and improved signal detection in the measurement of ctDNA/ctRNA or additional circulating tumor components such as CTCs. Nonetheless, these promising non-invasive biomarkers need to be validated in large cohorts of patients in order to reveal a clinical translation. Further research contemplating the combined analysis of circulating components should be take into account, since each has the potential to capture unique aspects of the characteristics of the tumor genome, and together might become the most reliable and comprehensive tool of information. 

Overall, liquid biopsy emerges as an attractive minimally invasive approach to identify diagnostic and prognostic biomarkers in biological fluids as well as a potential tool for monitoring tumor evolution, therapy response and treatment selection. Validation of the existing results is still needed and will strengthen the robustness of the data.

## Figures and Tables

**Figure 1 cancers-11-00950-f001:**
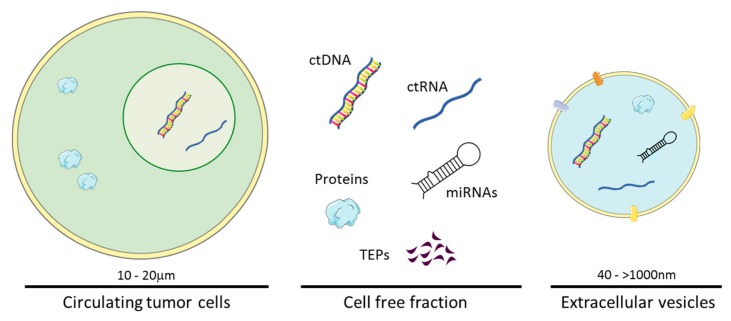
Components found in liquid biopsy (blood or cerebrospinal fluid) of glioblastoma patients. Liquid biopsies of tumor-specific circulating components include circulating tumor cells (CTCs), circulating cell-free tumor DNA (ctDNA) and RNA (ctRNA), circulating microRNAs (miRNAs), proteins, tumor-educated platelets (TEPs) and extracellular vesicles (EVs).

**Figure 2 cancers-11-00950-f002:**
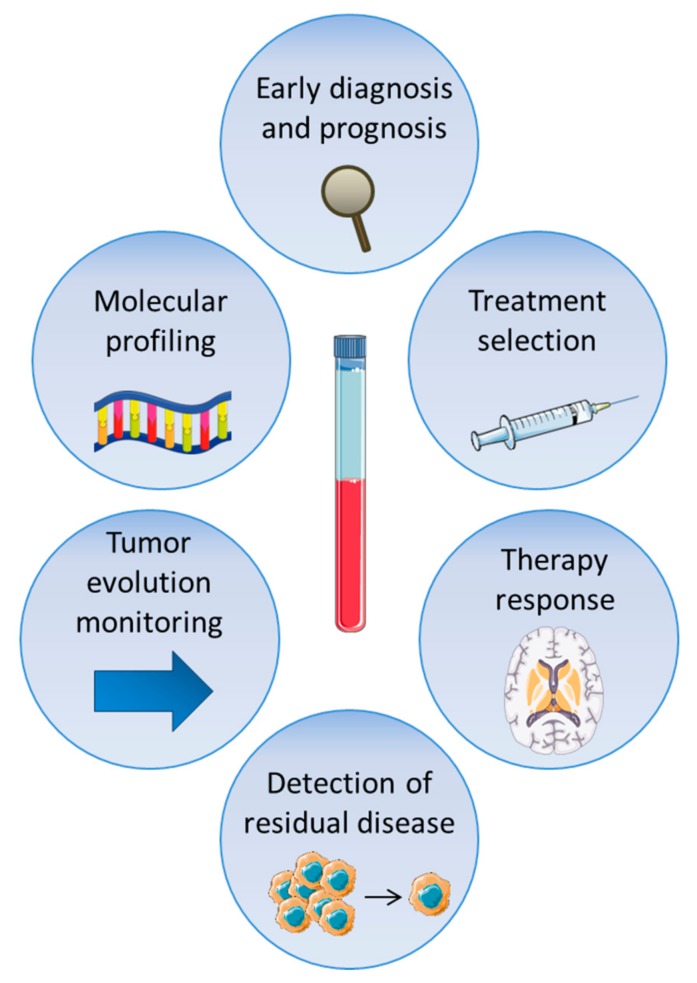
Potential applications of liquid biopsy in glioblastoma. The different circulating molecules may be applied to guide initial diagnosis, prognosis prediction, the molecular profiling of the tumor, treatment planning, patient follow-up, tumor evolution monitoring and the detection of glioblastoma relapse.

**Table 1 cancers-11-00950-t001:** Components identified in the blood of glioblastoma patients.

Liquid Biopsy Component	Molecule	Example	Information Provided and Findings	Clinical Applicability	Reference
**Cell-free components**	ctDNA	*MGMT*	promoter methylation status	therapy response	[21]
		*MGMT*	promoter methylation status	diagnosis	[22]
		*IDH1, EGFR, TP53, PTEN*	mutations	molecular profiling/diagnosis/prognosis	[16]
		*TP53, NF1, MET, APC, PDGFRA*	mutations	molecular profiling/treatment selection	[23]
		*MET, EGFR, ERBB2*	amplifications	molecular profiling/treatment selection	[23]
	miRNA	miR-15b	downregulated	diagnosis	[24,25]
		miR-23a, miR-133a, miR-150, miR-197, miR-548b	downregulated	diagnosis	[24]
		miR-128, miR-342	downregulated	therapy response/diagnosis	[25]
		miR-125b	downregulated	diagnosis	[26,27]
		miR-497	downregulated	diagnosis	[24,26]
		miR-205	downregulated	diagnosis/prognosis/tumor evolution monitoring	[28]
		miR-21	upregulated	diagnosis/prognosis/therapy response	[25,29]
		miR-221, miR-222	upregulated	prognosis	[30]
		miR-454	upregulated	prognosis	[31]
		miR-210, miR-182	upregulated	diagnosis	[32,33]
	lncRNA	HOTAIR	upregulated	diagnosis/prognosis	[34,35,36]
		GAS5	downregulated	prognosis/therapy response	[35]
		MALAT1, H19	deregulated	prognosis	[35]
	Protein	Haptoglobin α2	high levels	diagnosis/prognosis	[37]
		YKL-40	high levels	prognosis	[38,39]
		AHSG	high levels	prognosis	[40]
**EVs**	DNA	*EGFR*	amplification	diagnosis	[41]
		*PTEN*	deletion	diagnosis	[41]
		*TP53, IDH1/2*	mutations	diagnosis	[41]
		*EGFRvIII, IDH1, PTEN*	mutations	diagnosis/prognosis	[42,43]
		*MGMT*	high expression	therapy response	[44]
	Protein	PD-L1	surface expression	prognosis/therapy response	[45]
		TGM2, CD44, CD133	high levels	therapy response	[46]
**CTCs**	DNA	*SERPINE1, TGFB1, TGFBR2, VIM*	high expression	diagnosis/prognosis	[47]
		*EGFR*	amplification	prognosis	[48,49]
		*SOX2, OCT4, NANOG*	expression	prognosis	[50,51]
**TEPs**		*EGFRvIII*	mutation	diagnosis	[52]

Circulating tumor DNA (ctDNA), microRNA (miRNA), long non-coding RNA (lncRNA), extracellular vesicles (EVs), circulating tumor cells (CTCs), tumor-educated platelets (TEPs), O6-methylguanine-DNA-methyltransferase (*MGMT*), isocitrate dehydrogenase (*IDH*), epidermal growth factor receptor (*EGFR*), tumor protein p53 (*TP53*), phosphatase and tensin homolog (*PTEN*), neurofibromin 1 (*NF1*), MET proto-oncogene receptor tyrosine kinase (*MET*), APC regulator of Wnt signaling pathway (*APC*), platelet derived growth factor receptor alpha (*PDGFRA*), ErbB2 receptor tyrosine kinase 2 (*ERBB2*), HOX antisense intergenic RNA (HOTAIR), growth arrest-specific transcript 5 (GAS5), metastasis associated in lung adenocarcinoma transcript 1 (MALAT1), H19 imprinted maternally-expressed transcript (H19), epidermal growth factor receptor variant III (*EGFRvIII*), programmed death ligand 1 (PD-L1), transglutaminase 2 (TGM2), serpin family E member 1 (*SERPINE1*), transforming growth factor beta 1 (*TGFB1*), transforming growth factor beta receptor 2 (*TGFBR2*), vimentin (*VIM*), SRY (sex determining region Y)-box 2 (*SOX2*), octamer-binding transcription factor 4 (*OCT4*), and nanog homeobox (*NANOG*).

**Table 2 cancers-11-00950-t002:** Components identified in the cerebrospinal fluid (CSF) of glioblastoma patients.

Liquid Biopsy Component	Molecule	Example	Information Provided and Findings	Clinical Applicability	Reference
**Cell-free components**	ctDNA	*IDH1, TP53, PTEN, EGFR, FGFR2, ERBB2*	mutations	diagnosis/prognosis	[17]
		*IDH1, IDH2, TP53*	mutations	diagnosis/prognosis	[96]
		*CDKN2A, CDKN2B*	deletions	diagnosis/prognosis	[96]
		*EGFR*	amplification	diagnosis/prognosis	[96]
		*MGMT, p16INK4a, TIMP3, THBS1*	promoter hypermethylation	Diagnosis	[97]
	miRNA	miR125b, miR-223, miR-451, miR-711, miR-935	upregulated	diagnosis/therapy response	[98]
		miR-21, miR-218, miR-193b, miR-331, miR374a, miR548c, miR520f, miR27b and miR-30b	upregulated	diagnosis/therapy response	[99,100]
		miR-21, miR-10b	upregulated	diagnosis/prognosis/therapy response	[100,101,102,103]
		miR-15b	upregulated	prognosis/therapy response	[101]
	Protein	Tenascin	high levels	Diagnosis	[104]
		OPN	high levels	Diagnosis	[105]
		VEGF	high levels	diagnosis/prognosis	[106,107]
		TNFα, CCL3, CCL4	high levels	diagnosis/prognosis	[106]
		FGF2	high levels	diagnosis/prognosis	[107]
		NGF	high levels	prognosis/tumor progression	[108]
**EVs**	DNA	*IDH1*	mutation	Diagnosis	[109]
	miRNA	miR-21	upregulated	diagnosis/prognosis	[82]

Circulating tumor DNA (ctDNA), microRNA (miRNA), extracellular vesicles (EVs), isocitrate dehydrogenase (*IDH*), tumor protein p53 (*TP53*), phosphatase and tensin homolog (*PTEN*), epidermal growth factor receptor (*EGFR*), fibroblast growth factor receptor 2 (*FGFR2*), ErbB2 receptor tyrosine kinase 2 (*ERBB2*), cyclin dependent kinase inhibitor (*CDKN*), O6-methylguanine-DNA-methyltransferase (*MGMT*), TIMP metallopeptidase inhibitor 3 (*TIMP3*), thrombospondin 1 (*THBS1*), osteopontin (OPN), vascular endothelial growth factor (VEGF), tumor necrosis factor alpha (TNFα), C-C motif chemokine ligand (CCL), fibroblast growth factor 2 (FGF2), neural growth factor (NGF).

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
