# Peer review of "Liquid Biopsy in Glioblastoma: Opportunities, Applications and Challenges"

_cancers, 2019, doi:10.3390/cancers11070950_

Round 1
Reviewer 1 Report
The review gives a good overview over the different sources for liquid biopsies. By describing the different body fluids and biomolecules in great details, the specifics for glioblastoma liquid biopsy are lacking behind the general descriptions. Therefore it is difficult to actually get a good idea of what is possible with liquid biopsies in glioblastoma patients. A stronger emphasis on tumor specific findings might be beneficial. Even though progress has been made in the last years in this field, it is still not in clinical routines and this fact could be discussed in more detail.
Major criticism:
- The graphic indicates a number of applications (misspelling ‘aplications’) that are actually not discussed in the text, eg tumor evolution, treatment selection.
- In the paragraph about miRNA there is some discrepancy in the levels of certain miRNA line 182/183 ‘reduced serum levels of miR-15b which is the opposite of line 200 with elevated serum levels of miR15b. Similar in line 184 significant downregulation of miR-21, miR-128, miR-342-3p and line 198 the same miRNAs being highly expressed in high grade gliomas-
Minor criticism:
- CSF often misspelled as CFS
-in line 355 a primary cns in mentioned, probably meant is a primary CNS tumor
Author Response
The review gives a good overview over the different sources for liquid biopsies. By describing the different body fluids and biomolecules in great details, the specifics for glioblastoma liquid biopsy are lacking behind the general descriptions. Therefore it is difficult to actually get a good idea of what is possible with liquid biopsies in glioblastoma patients. A stronger emphasis on tumor specific findings might be beneficial. Even though progress has been made in the last years in this field, it is still not in clinical routines and this fact could be discussed in more detail.
AUTHORS: We thank the reviewer for the insightful suggestions. The manuscript has been significantly changed (indeed it changed so much that we have preferred not to include the modification with track changes) increasing the emphasis on the impact of liquid biopsies specifically in glioblastoma patients. For this, (i) we removed the section of introduction of Liquid biopsy and leave only the section of Liquid biopsy in glioblastoma. (ii) we excluded general comments and several cites that were not related to glioblastoma. (iii) we have added 2 new tables (Table 2 and 3) that extensively describe the main discoveries in glioblastoma in blood and CF respectively.
We also further discussed the hurdles for implementation of liquid biopsy in clinical practice in each section as well as in the conclusions section, which has been significantly changed. Moreover, we present a new table (Table 1) showing the main advantages and disadvantages of liquid biopsy compared to tissue sample.
Major criticism:
- The graphic indicates a number of applications (misspelling ‘aplications’) that are actually not discussed in the text, eg tumor evolution, treatment selection.
AUTHORS: We have done several changes in the graphic. First we have divided the original graph in two figures in which we present separately the components found in liquid biopsies and the applications it might have. Moreover, we have adjusted the information presented in the figures to what is discussed in the text.
- In the paragraph about miRNA there is some discrepancy in the levels of certain miRNA line 182/183 ‘reduced serum levels of miR-15b which is the opposite of line 200 with elevated serum levels of miR15b. Similar in line 184 significant downregulation of miR-21, miR-128, miR-342-3p and line 198 the same miRNAs being highly expressed in high grade gliomas-
AUTHORS: We have significantly modified the paragraph about miRNAs describing more accurately the expression levels and their impact as biomarkers of diagnosis and prognosis in glioblastoma. New information can be seen in Section Liquid biopsy components from blood (lines 164 to 2018) and Liquid biopsy in cerebrospinal fluid (lines 426 to 446).
Minor criticism:
- CSF often misspelled as CFS
-in line 355 a primary cns in mentioned, probably meant is a primary CNS tumor
AUTHORS: These errors have been corrected in the revised version of the manuscript.
Reviewer 2 Report
After a brief introduction on glioblastomas and liquid biopsies in general, the authors summarize in this review the components in liquid biopsies (mainly blood, but also CSF and urine) that might be used for assessment of diagnosis, prognosis and for improved therapeutic management. Although the review is quite systematically written, in its present form it suffers from several shortcomings that are listed below.
1. In the Title of the manuscript the authors ‘promise’ to discuss opportunities, applications an challenges of liquid biopsies in patients with glioblastoma. However, if one takes the title serious review is in my opinion not well-balanced as there is a lot of information on opportunities, but relatively little information on the applications (e.g. is it already used in clinical practice??) and on the challenges/hurdles in this context; a more systematic discussion of these challenges/hurdles that still need to be tackled before liquid biopsy diagnosis can be used in routine clinical practice would do the review good.
2. The authors suggest in this review that ‘ctDNA provides a comprehensive view of the tumor genome, as it reflects DNA derived from multiple tumor regions or metastasis’. However, it is already clear that different (e.g. more and less malignant) parts of the same tumor indeed result in ctDNA signals in liquid biopsies that can later on be discriminated again as being derived from different parts of/cells in the tumor?? In other words, will it indeed be possible to decipher the information in liquid biopsies in such a way that it will allow for reliable reconstruction of intratumoral heterogeneity at e.g. the DNA level?
3. The authors should be more precise and more consistent in their statements, e.g.:
- names of (human) genes should consequently be written in italics;
- in some places they use the abbreviation GBM, in other places they just write glioblastoma;
- page 3, line 95: ‘… MRI can only detect established tumors with sufficient mass.’; not sure if this is true, as very subtle lesions causing hardly any mass effect can still be seen on MRI due to disruption of the blood-brain barrier etc.;
- page 5, lines 191-194: ‘… a restricted signature of serum miRNAs … could distinguish high-grade and low-grade gliomas, whereas tissue samples only allow differentiating glioblastoma patients from normal controls’; how about that? I would think that based on combined histological and molecular analysis of glioma tissue should allow for pinpointing the nature of the lesion in the vast majority of cases;
- page 6, lines 236 and 252: ‘… exosomes arise from viable cancer cells …’ and ‘EV-derived DNA may be representative of the entire genome …’; I don’t understand right away why exosomes derived from viable cancer cells would contain DNA of these cells that is representative of the entire genome of these cells …;
- page 7, lines 282-283: ‘Measurements of circulating proteins can identify potential reliable biomarkers for different diseases.’; In fact, measurement of circulating proteins has already provided very useful/helpful information in diagnosing and monitoring a wide spectrum of diseases (not just cancer) for decades!
- Etc.
4. The authors refer in part 1 of this review to 4 subtypes of glioblastomas: classical, mesenchymal, proneural, neural. However, as far as I know the neural subtype is now kind of obsolete as this profile was for a large part due to admixture of non-neoplastic brain …
5. The paper still needs serious editing anyway as in its present form it still suffers from a multitude of (linguistic) flaws, e.g.:
- Abstract (line 20-21): This advances … (These advances);
- Abstract (line 22): … therapy guide. (guidance of therapy);
- Abstract (line 23): we will summarize … in gliomas, (but only glioblastomas are discussed in this review);
- Legend Fig. 1: ‘… miRNAs, circulating miRNAs …’ (some redundancy here);
- Etc. etc.
Author Response
After a brief introduction on glioblastomas and liquid biopsies in general, the authors summarize in this review the components in liquid biopsies (mainly blood, but also CSF and urine) that might be used for assessment of diagnosis, prognosis and for improved therapeutic management. Although the review is quite systematically written, in its present form it suffers from several shortcomings that are listed below.
1. In the Title of the manuscript the authors ‘promise’ to discuss opportunities, applications an challenges of liquid biopsies in patients with glioblastoma. However, if one takes the title serious review is in my opinion not well-balanced as there is a lot of information on opportunities, but relatively little information on the applications (e.g. is it already used in clinical practice??) and on the challenges/hurdles in this context; a more systematic discussion of these challenges/hurdles that still need to be tackled before liquid biopsy diagnosis can be used in routine clinical practice would do the review good.
AUTHORS: We thank the reviewer for the insightful suggestion, which is very similar to the observation of referee 1. Consequently, we have further discussed the hurdles for implementation of liquid biopsy in clinical practice in each section as well as in the conclusions section. Moreover, we have highlighted the relevant information in the new Tables 1 to 3.
2. The authors suggest in this review that ‘ctDNA provides a comprehensive view of the tumor genome, as it reflects DNA derived from multiple tumor regions or metastasis’. However, it is already clear that different (e.g. more and less malignant) parts of the same tumor indeed result in ctDNA signals in liquid biopsies that can later on be discriminated again as being derived from different parts of/cells in the tumor?? In other words, will it indeed be possible to decipher the information in liquid biopsies in such a way that it will allow for reliable reconstruction of intratumoral heterogeneity at e.g. the DNA level?
AUTHORS: We understand reviewer´s concern and we agree with him/her. It has been shown that components of liquid biopsy present genetic alterations that correlate with the observed in the tumor tissue samples but nowadays there is no data that supports that liquid biopsy might reconstruct intratumoral heterogeneity with more sensibility and specificity of tumor tissue samples. We have toned down our sentence describing the idea explained here.
3. The authors should be more precise and more consistent in their statements, e.g.names of (human) genes should consequently be written in italics;
-in some places they use the abbreviation GBM, in other places they just write glioblastoma;
-page 3, line 95: ‘… MRI can only detect established tumors with sufficient mass.’; not sure if this is true, as very subtle lesions causing hardly any mass effect can still be seen on MRI due to disruption of the blood-brain barrier etc.;
AUTHORS: We apologize for these errors that have been corrected in the revised version of the manuscript. We have paid significant attention in order to avoid typos, English errors, etc.
- page 5, lines 191-194: ‘… a restricted signature of serum miRNAs … could distinguish high-grade and low-grade gliomas, whereas tissue samples only allow differentiating glioblastoma patients from normal controls’; how about that? I would think that based on combined histological and molecular analysis of glioma tissue should allow for pinpointing the nature of the lesion in the vast majority of cases;
AUTHORS: We understand reviewer´s concern and we agree with him/her. It has been shown that components of liquid biopsy present genetic alterations that correlate with the observed in the tumor tissue samples but nowadays there is no data that supports that liquid biopsy might reconstruct intratumoral heterogeneity with more sensibility and specificity of tumor tissue samples. We have toned down our sentence describing the idea explained here.
The information regarding miRNAs is presented in (lines 164 to 2018) and (lines 426 to 446) for blood and CSF respectively.
- page 6, lines 236 and 252: ‘… exosomes arise from viable cancer cells …’ and ‘EV-derived DNA may be representative of the entire genome …’; I don’t understand right away why exosomes derived from viable cancer cells would contain DNA of these cells that is representative of the entire genome of these cells …;
AUTHORS: We have significantly changed the section of EV and exosomes being more precise and restrained in the description and interpretation of the results (Lines 310 to 336).
- page 7, lines 282-283: ‘Measurements of circulating proteins can identify potential reliable biomarkers for different diseases.’; In fact, measurement of circulating proteins has already provided very useful/helpful information in diagnosing and monitoring a wide spectrum of diseases (not just cancer) for decades!
AUTHORS: We wanted to highlight the positive impact that measurement of specific proteins could have in glioblastoma diagnosis and prognosis, but it is clear that we didn´t express it properly. This section has been significantly changed and improved following reviewer´s suggestion (Lines 310 to 336)
- Etc.
4. The authors refer in part 1 of this review to 4 subtypes of glioblastomas: classical, mesenchymal, proneural, neural. However, as far as I know the neural subtype is now kind of obsolete as this profile was for a large part due to admixture of non-neoplastic brain …
AUTHORS: The reviewer is right and we have corrected this point in the revised version of the review.
5. The paper still needs serious editing anyway as in its present form it still suffers from a multitude of (linguistic) flaws, e.g.: Abstract (line 20-21): This advances … (These advances); Abstract (line 22): … therapy guide. (guidance of therapy); Abstract (line 23): we will summarize … in gliomas, (but only glioblastomas are discussed in this review); Legend Fig. 1: miRNAs, circulating miRNAs …’ (some redundancy here); Etc. etc.
AUTHORS: As indicated above, we apologize for the errors that we have corrected in the revised version.
Reviewer 3 Report
Saenz et al wrote a concise review of recent literature on the concept of liquid biopsy in neuro-oncology, particularly regarding glioblastoma. The main sources of somatic informations are discussed, namely blood and CSF. Some major and minor comments are presented.
Major comments :
1/ The authors give too much importance to the plasma in glioblastoma with regard to the divergent results in the literature, especially on the detection ratio of somatic mutations on ctDNA. Reference 26 (Piccioni et al CNS Oncol 2019) is highly questionable. Their results should be more discussed, especially with regard to the lower detection rates previously published (ex: Juratli et al, Clin Cancer Research 2018). Non-confirmation by other sequencing techniques (eg, digital PCR) is an important limitation of their results. Barault et al (reference 28) did not identify MGMTp methylation in blood from GBM patients but from CCRm patients.
2/ The results on the LCR are not sufficiently developed and should be a chapter alone. The authors should also present concordance or discordance results on DNA sequencing between primary tumor and ctDNA.
Minor comments :
1/ CSF instead of CFS in 5.1.
2/ Attention to confusions between cfDNA and ctDNA.
Author Response
Saenz et al wrote a concise review of recent literature on the concept of liquid biopsy in neuro-oncology, particularly regarding glioblastoma. The main sources of somatic informations are discussed, namely blood and CSF. Some major and minor comments are presented.
Major comments :
1/ The authors give too much importance to the plasma in glioblastoma with regard to the divergent results in the literature, especially on the detection ratio of somatic mutations on ctDNA.Reference 26 (Piccioni et al CNS Oncol 2019) is highly questionable. Their results should be more discussed, especially with regard to the lower detection rates previously published (ex: Juratli et al, Clin Cancer Research 2018). Non-confirmation by other sequencing techniques (eg, digital PCR) is an important limitation of their results. Barault et al (reference 28) did not identify MGMTp methylation in blood from GBM patients but from CCRm patients.
AUTHORS :We thank the reviewer for this overview. We have done several points to address the reviewer´s comments: (i) we have discussed more in detail the techniques and results obtained in the manuscripts cited by the reviewer. (ii) we have extended the information regarding the strengths and limitations of the different techniques used to detect ctDNA. These points are presented from lines 93 to 148 of the revised version (iii) we have highlighted and extended the results obtained in CSF as an alternative and more sensitive tool.
2/ The results on the LCR are not sufficiently developed and should be a chapter alone. The authors should also present concordance or discordance results on DNA sequencing between primary tumor and ctDNA.
AUTHORS : The results obtained in CSF have been discussed more deeply in the revised version of the review. Moreover, their relevance has been highlighted as they were presented in a specific section (Section 4, line 366 to 441)) and also in a specific table (Table 3).
Minor comments :
1/ CSF instead of CFS in 5.1.
2/ Attention to confusions between cfDNA and ctDNA.
AUTHORS: We apologize for these errors that have been corrected in the revised version of the manuscript.
Reviewer 4 Report
In this review, the authors describe the different kind of circulating biomarkers and describe the liquid biopsy applications in glioblastoma. The article is divided into six sections, the two first ones being two introductions on glioblastoma and liquid biopsy, respectively. The third section appears as a short paragraph, listing the potential applications of liquid biopsy in light of the strides in tumor profiling and tumor tracking in glioblastoma. The fourth part is subdivided into six paragraphs describing the different types of circulating biomarkers (what the authors called "liquid biopsy components"). A fifth section addresses the question of the use of other sources for liquid biopsy such as cerebrospinal fluid and urine. Finally, the authors approach potential future directions in a conclusion.
Overall, this article presents a number of shortcomings in both form and substance.
The quality of presentation could be significantly improved by reconsidering the plan of the article. As presently drafted, there are two general introductions, one on glioblastoma and one on liquid biopsy and the aim of the review as well as the plan are not clearly announced. The glioblastoma introduction should be reworded in a more concise way. The way it is currently written is not in adequacy with the data of the literature. The flow and the size of each paragraphs should be proportional to the current knowledge and evidence-based to emphasize the most scientifically advanced and proved data. Thus, the part about circulating tumor DNA should be more developed than the other circulating biomarkers such as cfRNAs or extracellular vesicles. Taking in account the promising recent published results on the use of CSF as source of circulating tumor DNA in brain tumors, the authors should discuss more in detail these data and expand their bibliography that is currently incomplete. As it stands, the review is superficial and lack of comprehensiveness. Technologic pitfalls are not addressed. In order to make it more original, it would be worthwhile to discuss the different methodologies used to detect ctDNA, further list benefits and drawbacks of each, to reason about the positivity threshold and false positivity, ... .
Furthermore, the review presents shortcuts and approximations.
Figure 1 is superfluous. The text is unreadable (font size). The title is inappropriate; what do the authors mean here by "molecular heterogeneity"?
Need some English rewording and checking for typos.
Overall, the original contribution of this review is limited compared to the previous published articles on this topic.
Author Response
In this review, the authors describe the different kind of circulating biomarkers and describe the liquid biopsy applications in glioblastoma. The article is divided into six sections, the two first ones being two introductions on glioblastoma and liquid biopsy, respectively. The third section appears as a short paragraph, listing the potential applications of liquid biopsy in light of the strides in tumor profiling and tumor tracking in glioblastoma. The fourth part is subdivided into six paragraphs describing the different types of circulating biomarkers (what the authors called "liquid biopsy components"). A fifth section addresses the question of the use of other sources for liquid biopsy such as cerebrospinal fluid and urine. Finally, the authors approach potential future directions in a conclusion.
Overall, this article presents a number of shortcomings in both form and substance.
The quality of presentation could be significantly improved by reconsidering the plan of the article. As presently drafted, there are two general introductions, one on glioblastoma and one on liquid biopsy and the aim of the review as well as the plan are not clearly announced. The glioblastoma introduction should be reworded in a more concise way. The way it is currently written is not in adequacy with the data of the literature.
AUTHORS We thank the reviewer for suggesting an additional structure for our review, which we have followed. Thus, the revised version of the review presents a shorter introduction of glioblastoma (lines 3 to 45) and second section already presenting the benefits and some information of liquid biopsy in glioblastoma (lines 48 to 88).
The flow and the size of each paragraphs should be proportional to the current knowledge and evidence-based to emphasize the most scientifically advanced and proved data. Thus, the part about circulating tumor DNA should be more developed than the other circulating biomarkers such as cfRNAs or extracellular vesicles.
AUTHORS We thank the reviewer for this suggestion, which we have followed. Thus, the revised version of the review presents a much larger and detailed part of circulating tumor DNA in blood section (lines 93 to 158) and also in CSF (lines 388 to 422).
Taking in account the promising recent published results on the use of CSF as source of circulating tumor DNA in brain tumors, the authors should discuss more in detail these data and expand their bibliography that is currently incomplete. As it stands, the review is superficial and lack of comprehensiveness. Technologic pitfalls are not addressed. In order to make it more original, it would be worthwhile to discuss the different methodologies used to detect ctDNA, further list benefits and drawbacks of each, to reason about the positivity threshold and false positivity, ... .
AUTHORS : The part of CSF has been also significantly expanded becoming an independent section in the revised version of the review (section 4), with extended information regarding results and also technological pitfalls (lines 368 to 551). In addition, we present a new table describing the main data obtained in CSF (Table 3)
Furthermore, the review presents shortcuts and approximations.
Figure 1 is superfluous. The text is unreadable (font size). The title is inappropriate; what do the authors mean here by "molecular heterogeneity"?
AUTHORS : Following reviewer´s suggestion Figure 1 has been removed and we present 2 completely re-structured figures showing
Figure 1 - components found in liquid biopsy (blood or CSF) and,
Figure 2 applications of liquid biopsy in glioblastoma
Need some English rewording and checking for typos.
We apologize for these errors that have been corrected in the revised version of the manuscript. We have paid significant attention in order to avoid typos, English errors, etc.
Overall, the original contribution of this review is limited compared to the previous published articles on this topic.
We have massively reformatted the structure and content of the review and we hope that the presented revised version increases significantly reviewer´s opinion and considers our work suitable for publication in Cancers.
Round 2
Reviewer 2 Report
I’ve now read the reply of the authors to my critical comments and the changes the authors have made in the revised version of the manuscript (highlighted in yellow). In my opinion this revised manuscript is indeed an improvement. However: 1. Regarding the multitude of linquistic flaws the authors state that they ‘apologize for these errors that have been corrected in the revised version of the manuscript’ and that they ‘have paid significant attention in order to avoid typos, English errors, etc.’; unfortunately, I still see several errors in the revised manuscript as well, here some examples: * Line 85: ‘is not easy and feasible’; some redundancy here; in a way not feasible is an extreme form of not possible …; * Line 141: ‘they only enables’; should be: they only enable; * Line 144: ‘sensitive method to screen IDH1 mutations’; (screen for); * Line 367 & 369: OPN vs ONP …; * Line 404/405: ‘in all glioblastoma patients but in any of the healthy individuals’; not in any of the healthy individuals? * Line 418: upregulated/deregulated; do the authors really mean ‘deregulated’ here? * Etc. In other words, the authors still should do a better job to get the language of the manuscript right! 2. 2. In their reply to my second comment the authors state that ‘there is no data that supports that liquid biopsy might reconstruct intratumoral heterogeneity with more sensibility (I guess the authors mean sensitivity here) and specificity of tumor tissue samples’; still, in Table 1 in the revised version of the manuscript the authors state under Liquid biopsy:‘Captures molecular heterogeneity’ … 3. 3. The information provided by Table 2 and 3 is helpful, but not consistent throughout the Tables, e.g. * MGMT promoter methylation status (as in Table 3) or just methylation status (as in Table 2)? * And MGMT promoter methylation status informative for therapy respons and diagnosis (as in Table 2) or just for diagnosis (as in Table 3)? * What does ‘deregulated’ mean in Table 2 (after MALAT1, H19)? The authors should check these tables again for consistency and accuracy.Author Response
The text and tables have been checked and corrected following referee´s suggestions
Reviewer 3 Report
Saenz et al took into consideration the comments made at first reading and responded satisfactorily. The proposed review warrat publication in Cancers after a minor revision :
- italicize MGMT and THBS1 line 407.
Author Response
We have done the suggested change in the text
Reviewer 4 Report
Authors have carefully taken note of the comments made in the first review. In this revised manuscript, authors made significant efforts to reply point by point and refinements have been made.
Overall, the authors make overstatements and should tone down their claims in the conclusion. While some data are promising, they should temper their remarks. For example, one of the most recent published work on cfDNA, which is cited by the authors (Miller et al., ref 102), reported that mutations were detected only in half of the analyzed CSF while 19 plasmas analyzed allowed detection of mutations in only 3 samples. The analyses were performed in patients with advanced disease (disease duration at sampling was 463 days that is quite a long time for such agressive tumors), never at diagnosis and all patients received radiotherapy (impact on the blood-brain barrier). Similar results were found in other studies such as Piccioni (ref27), Pan et al. and Martinez et al. (non cited by the authors). Thus, in spite of progress made in recent years, in the light of the divergent results in the literature, the low detection rates reported in several studies (lack of false positive evaluation particularly when a pre-amplification step is performed), etc..., the clinical applications are still questionnable.
The manuscript should be further improved in that direction.
- Table one is not essential, unclear (positive and negative points are mixed) and contains incorrect statements (for example: why "no biomarker application"?). this table has to be removed.
- Figure 2:
Typos in the title and the legend
Please precise the title : 'Potential applications'
Author Response
We have toned down the claims in the conclusion section and removed the table 1 from the text.
We have also modified the figure 2 as suggested by the reviewer.